

# Analysis of protein-heparin interactions using a portable SPR instrument

Dunhao Su[1,*], Yong Li[1,*], Edwin A. Yates[1], Mark A. Skidmore[2], Marcelo A. Lima[2] and David G. Fernig[1]

[1] Biochemistry, University of Liverpool, Liverpool, United Kingdom
[2] Molecular & Structural Biosciences, School of Life Sciences, University of Keele, Newcastle-Under-Lyme, United Kingdom
* These authors contributed equally to this work.

## ABSTRACT

Optical biosensors such as those based on surface plasmon resonance (SPR) are a key analytical tool for understanding biomolecular interactions and function as well as the quantitative analysis of analytes in a wide variety of settings. The advent of portable SPR instruments enables analyses in the field. A critical step in method development is the passivation and functionalisation of the sensor surface. We describe the assembly of a surface of thiolated oleyl ethylene glycol/biotin oleyl ethylene glycol and its functionalisation with streptavidin and reducing end biotinylated heparin for a portable SPR instrument. Such surfaces can be batch prepared and stored. Two examples of the analysis of heparin-binding proteins are presented. The binding of fibroblast growth factor 2 and competition for the binding of a heparan sulfate sulfotransferase by a library of selectively modified heparins and suramin, which identify the selectivity of the enzyme for sulfated structures in the polysaccharide and demonstrate suramin as a competitor for the enzyme's sugar acceptor site. Heparin functionalised surfaces should have a wide applicability, since this polysaccharide is a close structural analogue of the host cell surface polysaccharide, heparan sulfate, a receptor for many endogenous proteins and viruses.

## INTRODUCTION

The regulation of biochemical processes depends on the dynamic interactions of biological molecules. The characterisation of these interactions represents an important step in gaining an understanding of function and how this may be perturbed in a biological experiment. Moreover, the ability of biological molecules such as antibodies and receptors, as well as various host entities, to recognise molecular partners with high selectivity has become integral to (bio)analytical techniques. A large number of different surface measurement platforms have been adapted for such analyses and of these, surface plasmon resonance and related techniques are some of the most popular, perhaps due to their early commercialisation (*Brigham-Burke, Edwards & O'Shannessy, 1992*; *Watts, Lowe &*

Corresponding author
David G. Fernig, dgfernig@liv.ac.uk

*Pollard-Knight, 1994*), rapidity and the ease of measurement. The technique has since evolved to encompass both analysis of functional molecular interactions and quantification of the amount of an analyte in a sample. The recent development of portable, low-cost instrumentation (*e.g.*, *Zhao et al., 2015*) has substantially increased the reach of the technique. In an SPR measurement, the ligand is immobilised on the sensor surface and the analyte is usually flowed over the sensor, though some systems do not use fluidics. The instruments use changes in the refractive index near the surface to measure in real-time the interactions of analyte with ligand.

Like any analytical method, SPR has limitations. One relates to the surface itself, while fluidics systems, which are characterised by inefficient exchange between the bulk, flowing liquid and the stationary layer, are a source of further artefacts. These have been reviewed extensively (*e.g.*, *Schuck & Zhao, 2010*). Another limitation relates to signal and noise. The principal source of noise is the interaction of the analyte with the surface to which the ligand is immobilised. Thus, an important part of experimental design is the synthesis of surfaces suitable for capturing ligands that exhibit very low analyte binding. Gold binds many groups that include thiols, amines and carbonyls, so has to be passivated. Dextran-based hydrogels were an early solution, but these are prone to high non-specific binding, as well as to generating artefacts arising from the collapse or expansion of the hydrogel upon interaction with an analyte (*Schuck & Zhao, 2010*). Since then, substantial advances involving useful cross-fertilisation between nanoparticle and surface sensing fields have been made in passivating gold surfaces with self-assembled monolayers (SAMs). These approaches include mercaptoproprionic acid (*Mauriz et al., 2006*), supported lipid bilayers (*Ferhan et al., 2019*; *Marques, de Almeida & Viana, 2014*), peptides (*Bolduc et al., 2009*; *Bolduc et al., 2011*; *Bolduc, Pelletier & Masson, 2010*) and thiolated oleyl ethylene glycol (OEG) (*Migliorini et al., 2014*). SAMs have many advantages; in particular, functionalisation can be achieved in the same step as passivation by the inclusion of a mole percent of a functional ligand. This allows statistical control of the functionalisation of the SAM and thereby the surface. Control of the surface density of functionalisation provides a means by which steric hindrance artefacts can be avoided (*Edwards et al., 1995*). Since the SAMs consist of small molecules, the immobilised ligand will also be close to the surface. Thus, a SAM affords greater sensitivity than a hydrogel, because the SPR signal decreases exponentially with the distance from the surface. No passivation system offers a universal panacea, however, so having access to several distinct SAMs increases the probability of finding a surface suitable for a particular measurement.

We have adapted an existing strategy for the functionalisation of gold surfaces with a monolayer of streptavidin that is particularly resistant to non-specific binding (*Migliorini et al., 2014*) to the SPR chips of a portable SPR instrument (*Zhao et al., 2015*). Self-assembled monolayers of defined mixtures of thiolated oleyl ethylene glycol (OEG) and biotin OEG were synthesised on the gold surface and used to capture streptavidin. Reducing-end biotinylated heparin (*Thakar et al., 2014*) was then captured on the streptavidin surface. Heparin is a convenient experimental proxy for the sulfated domains of cellular heparan sulfate, which by virtue of the latter's more than 800 extracellular

Table 1 Nomenclature of chemically modified heparin structures.

| Analogue | Predominant repeat | IdoA-2 | GlcN-6 | GlcN-2 | IdoA-3 | GlcN-3 |
|---|---|---|---|---|---|---|
| 1(heparin) | $I_{2S}A^{6S}_{NS}$ | $SO_3^-$ | $SO_3^-$ | $SO_3^-$ | OH | OH |
| 2 | $I_{2S}A^{6S}_{NAc}$ | $SO_3^-$ | $SO_3^-$ | $COCH_3$ | OH | OH |
| 3 | $I_{2OH}A^{6S}_{NS}$ | OH | $SO_3^-$ | $SO_3^-$ | OH | OH |
| 4 | $I_{2S}A^{6OH}_{NS}$ | $SO_3^-$ | OH | $SO_3^-$ | OH | OH |
| 5 | $I_{2OH}A^{6S}_{NAc}$ | OH | $SO_3^-$ | $COCH_3$ | OH | OH |
| 6 | $I_{2S}A^{6OH}_{NAc}$ | $SO_3^-$ | OH | $COCH_3$ | OH | OH |
| 7 | $I_{2OH}A^{6OH}_{NS}$ | OH | OH | $SO_3^-$ | OH | OH |
| 8 | $I_{2OH}A^{6OH}_{NAc}$ | OH | OH | $COCH_3$ | OH | OH |
| 9 | $I_{2S,3S}A^{6S}_{3S,NS}$ | $SO_3^-$ | $SO_3^-$ | $SO_3^-$ | $SO_3^-$ | $SO_3^-$ |

Note:
I denotes L-iduronate (IdoA), and A stands for the amino sugar, D-glucosamine (GlcN). Numbers refer to the ring position of the carbon atoms, OH denotes free hydroxyl, S, sulfate ester and NAc or NS represent N-acetylation of N-sulfation of the amino group of glucosamine, respectively.

protein partners is a major regulator of cell function in development, homeostasis and disease (Nunes et al., 2019). Indeed, this surface has recently been used to characterise the interaction of the spike protein of SARS-CoV-2 with heparin (Mycroft-West et al., 2020). Here we provide an in-depth description of the preparation of this surface, benchmark it against the well-characterised interaction of fibroblast growth factor 2 (FGF2) with heparin (Rahmoune et al., 1998) and demonstrate its application in the characterisation of heparan sulfate sulfotransferases, in this instance a fusion protein of glutathione-S-transferase (GST) and 3-O heparan sulfate sulfotransferase 3A1 (HS3ST3A1). We demonstrate that suramin, a previously characterised inhibitor of HS2ST1 that was postulated to compete for both the donor and acceptor sites of the enzyme (Byrne et al., 2018b), did indeed compete with GST-HS3ST3A1 binding to heparin. Additionally, the OEG/biotin OEG surfaces can be stored and re-used, and complement the existing peptide surfaces (Bolduc et al., 2009; Bolduc et al., 2011; Bolduc, Pelletier & Masson, 2010) that have been published for this portable SPR instrument.

# MATERIALS AND METHODS

## Materials

The chemically modified heparins (Table 1) were produced and characterised as described (Yates et al., 1996). Human fibroblast growth factor 2 (FGF2) and was produced as described by Duchesne et al. (2012).

## Oxime biotinylation of heparin at the reducing end

Porcine intestinal heparin (H4784; Sigma, Kanagawa, Japan) was biotinylated at the reducing end using hydroxylamine biotin, as described by Thakar et al. (2014), as this provides for a stable product. Heparin (4 mM) was reacted with 4 mM N-(aminooxyacetyl)-N′-(D-Biotinoyl) hydrazine, (A10550; ThermoFisher Scientific, Gloucester, UK) in 100 mM aniline and 100 mM acetate buffer, pH4.6, total volume 120 µl, at 37 °C for 48 h. After, unreacted biotin was removed and the buffer exchanged to

PBS (Thermo Scientific Oxoid, phosphate-buffered saline tablets, BR0014G) by six rounds of filtration at 10,000 g for 10 min in a 3 k MWCO centrifugal filter (88512; ThermoFisher Scientific, Gloucester, UK). The concentration of biotin-heparin was then measured using the carbazole assay. Briefly, 50 μL of 10-fold dilution of the biotin-heparin and a range of heparin standards at known concentration were added to a 96-well-plate, followed by the addition of 200 μL $H_2SO_4$ and then incubated at 98 °C for 20 min. After cooling to room temperature over 10 min, 20 μL of 1.25 mg/mL carbazole was added to the wells, which were then incubated at 98 °C for 10 min. After cooling to room temperature for 10 min the absorbance in the wells was measured at 535 nm, with the heparin standard curve enabling quantification of the biotin-heparin.

## Expression and purification of GST-HS3ST3A1

A cDNA fragment encoding the catalytic domain of human HS3ST3A1 (Uniprot accession Q9Y663, residues 139-406; purchased from Thermo Fisher, Gloucester, UK) was cloned into the pGEX4T3 vector using EcoRI and NotI. Glutathione-S-transferase-(GST) HS3ST3A1 was expressed and purified following published procedures for GST-HS3ST3A1 (*Moon et al., 2004*) and GST-HS3ST1 (*Wheeler et al., 2021*) fusion proteins. Protein was expressed in C41 (DE3) *E. coli*, induced with 200 μM isopropyl 1-thio-ß-D-galactopyranoside (IPTG) at 22 °C overnight. Following centrifugation (15 min, 4,150× g) and resuspension in lysis buffer (100 mM NaCl, 50 mM Tris-Cl, pH 7.4) cells were lysed by sonication (six 1 min 30 s cycles of sonication on ice) and the lysate cleared by centrifugation (45 min, 38,000 g). Soluble GST-HS3ST3A1 in the supernatant was purified by application to a 2 mL glutathione resin (Genscript Biotech Corporation, Rijswijk, Netherlands), washing with 100 mM NaCl, 50 mM Tris, pH 7.4 and eluting with 10 mM reduced glutathione, 100 mM NaCl, 50 mM Tris, pH 7.4. The eluate was immediately applied to a 1 mL heparin (Affi-Gel hep; Bio-Rad, Watford, UK) column, washed with 50 mM NaCl, 50 mM Tris, pH 7.4, and eluted with 600 mM NaCl, 50 mM Tris, pH 7.4 (Fig. S1). The specific extinction coefficient for GST-HS3ST3A1 at 280 nm, calculated from its amino acid sequence was used for quantification and protein was snap frozen in liquid nitrogen and stored in aliquots at −80 °C.

## Preparation of functionalised gold surfaces

The functionalisation of the gold sensor surface was based on an existing method (*Migliorini et al., 2014*). The detailed protocol for this is available at protocols.io (*Su & Fernig, 2021*). Briefly, mixtures of thiolated oleyl ethylene glycol (OEG) and thiolated oleyl ethylene glycol-biotin (OEG-biotin) at the stated % mole/mole and at a final concentration of 100 mM were prepared in ethanol. A plasma cleaned gold SPR sensor chip was incubated in this solution for 36 h to form an OEG/OEG-biotin SAM. The sensor chip was washed in ethanol and could be stored at 4 °C in ethanol for at least 3 months. The sensor chip placed in the P4SPR, multi-channel SPR instrument (Affinté Instruments, Montréal, Canada) and the three measurement channels and the control, background channel, were then equilibrated in PBS at a flow rate of 500 μL/min using an Ismatec peristaltic pump. A high flow rate was chosen to minimize mass transport limitations

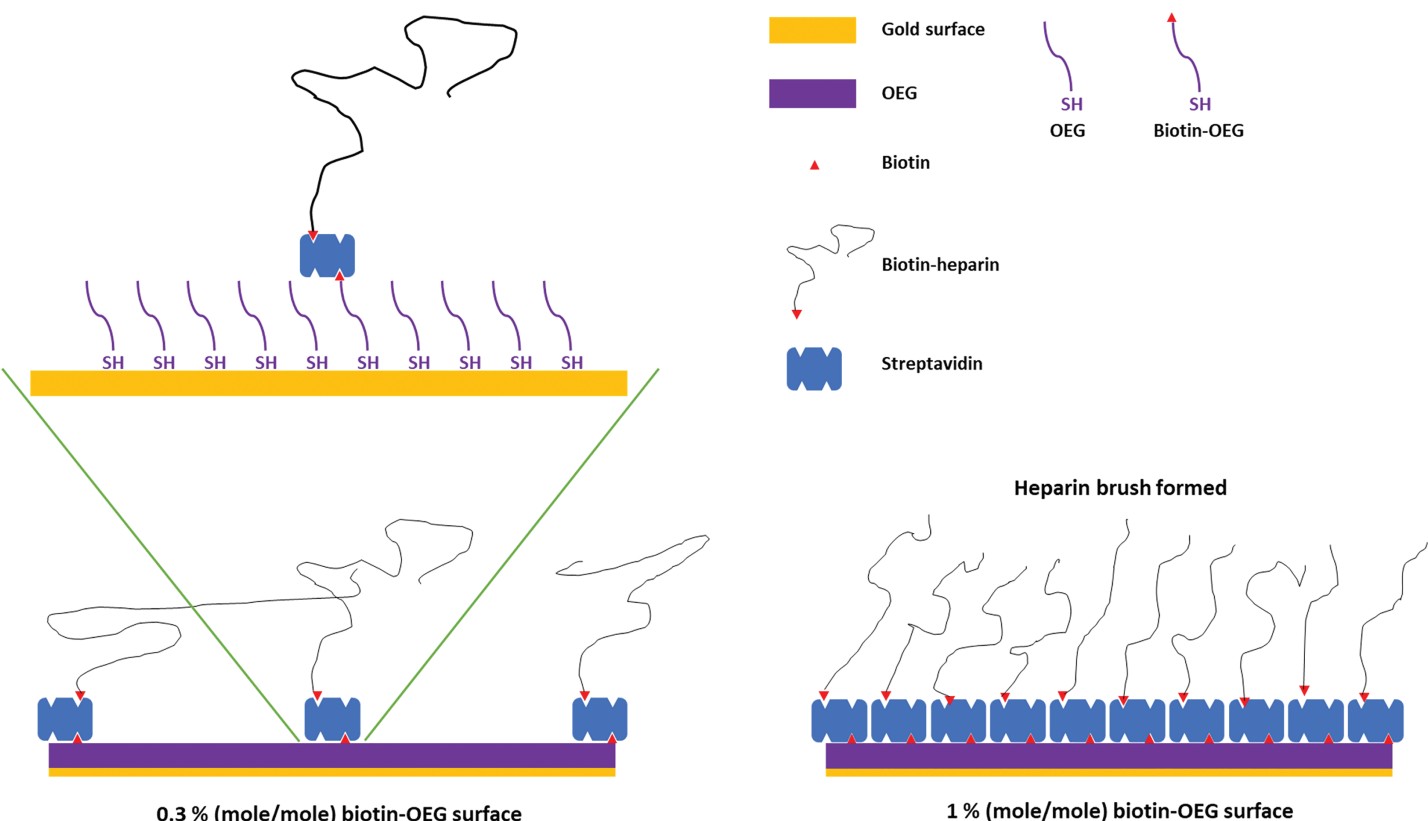

**Figure 1 Schematic of the assembly of the OEG self-assembled monolayer and subsequent capture of streptavidin and biotinylated ligand.** Left, illustration of the effect of reducing the mole percentage biotin OEG in the self-assembled monolayer on the density of streptavidin compared to (right) the densely packed streptavidin layer formed on a 1% (mole/mole) biotin OEG self-assembled monolayer. The latter results in the formation of a 'brush' of an end-labelled polymer, illustrated with reducing end biotinylated heparin.

(*Schuck & Zhao, 2010*), which are particularly acute when molecular interactions are driven by electrostatic interactions, as for proteins binding heparin, though lower flow rates are often appropriate, depending on the characteristics of the interaction and the assay. All subsequent steps used this flow rate. After equilibration in PBS, streptavidin (Sigma, Kanagawa, Japan), 1 mg/mL in PBS was injected over the four channels. Often a second injection of streptavidin was performed to determine whether the surfaces were fully functionalised. Streptavidin not specially bound to biotin was removed by two regeneration steps, the first using 2 M NaCl, the second 20 mM HCl. Then 1 mL, 90 μg/mL biotinylated heparin was injected over the three measurement channels. Again, a repeat injection of biotinylated heparin was performed to see if the streptavidin in the three measurement channels was fully derivatised. Finally, the three measurement channels were regenerated by sequential 1 mL injections of 2 M NaCl and 20 mM HCl (Fig. 1). The heparin functionalised surfaces could be stored in PBS at 4 °C for at least 10 days.

## Measurement of protein-heparin interactions

All measurements of the binding of protein analytes to immobilised heparin used PBS supplemented with 0.01% (v/v) Tween 20 (PBST) as the running buffer, to prevent

adsorption of analyte to the fluidics, at a flow rate of 500 µL/min. FGF2 and HS3ST3A were injected over the three measurement channels and the control channel at the concentrations indicated in the figure legends. The biotin-heparin surfaces were regenerated with 1 mL injections of 2 M NaCl and 20 mM HCl or 0.25% (w/v) SDS and 20 mM HCl. In some experiments the protein analyte was pre-mixed with a potential competitor for its binding to heparin prior to injection.

## RESULTS AND DISCUSSION

SAMs of thiolated OEG:thiolated biotin OEG functionalised with streptavidin on gold have been shown to be very resistant to non-specific binding and have been extensively characterised in terms of areal density of immobilized streptavidin and biotinylated polysaccharides (*Migliorini et al., 2014*). This approach also allows control of the surface density of streptavidin and hence biotinylated ligand immobilised on the surface by simply controlling the molar stoichiometry of thiolated OEG: thiolated biotin OEG (Fig. 1). Moreover, by capturing a biotinylated ligand, its orientation can be pre-determined if the biotinylation reaction is selective. Combined, these features reduce the likelihood of artefacts associated with diffusion in fluidics systems (*Schuck & Zhao, 2010*) and steric hindrance of the binding site(s) of the immobilised ligand (*Edwards et al., 1995*). Three (mole/mole) ratios were chosen, 1%, 0.3% and 0.1%. Assuming equal probability of incorporation of the thiolated biotin OEG into the monolayer and a diameter of streptavidin of ~5.5 nm, these should yield, respectively, a very tightly packed monolayer, a monolayer, and partial surface coverage of streptavidin.

Following plasma cleaning and incubation of the gold surfaces with the thiolated OEG/thiolated biotin OEG mixture for 36 h, the surfaces were washed with ethanol. The OEG/biotin-OEG SAMS were found to be stable for at least three months at 4 °C, which enabled batch preparation of surfaces. The thiolated OEG surface was placed in the instrument and equilibrated in running buffer (PBS, 500 µL/min), and then 1 mL streptavidin (20 µg/mL in PBS) was injected at the same flow rate providing a final response of ~2 nm (Fig. 2A). The surface was then subjected to a wash with 2 M NaCl and then 20 mM HCl (1 mL each) to remove any non-specifically bound streptavidin. A second injection of streptavidin only resulted in a very small increase in bound protein (Fig. 2A, red arrows, 800 min to 1,100 min), indicating that the biotinylated OEG on the surface was near saturated with streptavidin after the first injection. Nevertheless, all subsequent surfaces were prepared with two sequential injections of streptavidin, to ensure that this was indeed the case. The surface was again washed sequentially with 2 M NaCl and 20 mM HCl. Biotinylated heparin (1 mL, 90 µg/mL) was then injected over the three measurement channels, A–C. Only a small response was observed, as the refractive index of this sulfated polysaccharide is at the lower limit of detection (Fig. 2B red arrows 1,850 min to 2,000 min). After washing the surface with 2 M NaCl and 20 mM HCl, biotin heparin was again injected over the three measurement channels, A–C, but this did not cause any further increase in signal, demonstrating that the first single injection was sufficient to occupy the available sites on the immobilised streptavidin. Following washes with 2 M NaCl and 20 mM HCl, the surface was returned to running buffer.

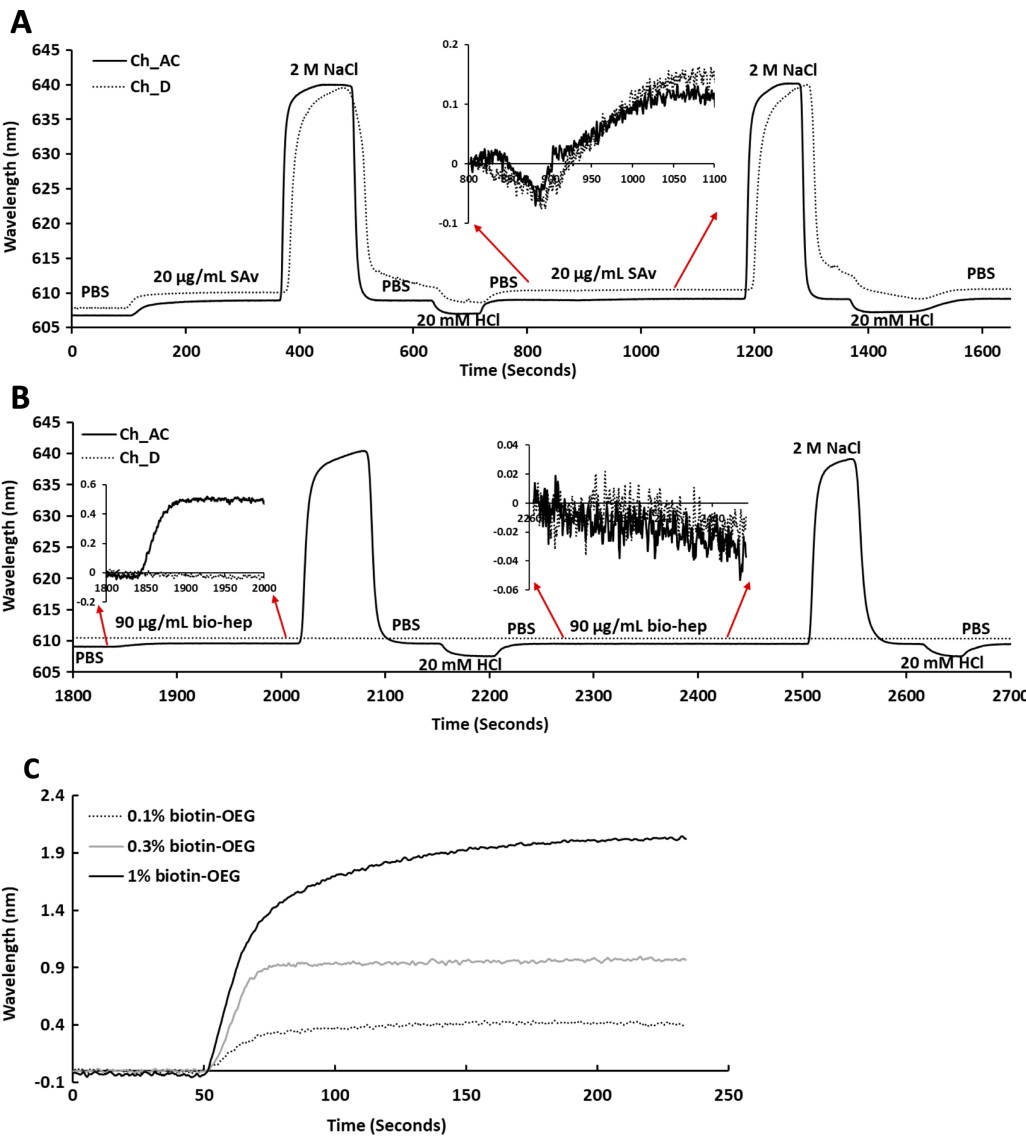

**Figure 2 Capture of streptavidin on biotin OEG/OEG self-assembled monolayers and functionalisation with reducing end biotinylated heparin.** Following *ex-situ* assembly of the biotin OEG/OEG monolayer, the chip was inserted into the P4SPR instrument and a flow rate of 500 µL/min PBS was maintained. (A) One mL of 20 µg/mL streptavidin in PBS was injected over a 1% (mole/mole) biotin OEG/OEG SAM in channels A–C and D, after which unbound streptavidin was removed by a 1 mL wash with 2 M NaCl in 10 mM (buffer), pH 7.2, and, after returning the surface to PBS with 1 mL 20 mM HCl. A repeat injection of 1 mL 20 µg/mL streptavidin in PBS over channels A–C and D (800 min, red arrows, inset) did not result in the capture of any further streptavidin on the surface. (B) Reducing end biotinylated heparin (90 µg/mL) was injected over the streptavidin derivatised a 1% (mole/mole) biotin OEG SAM in channels A–C from panel (A). After washing the surfaces with 2 M NaCl and 20 mm HCl to remove any loosely bound biotin heparin, a second injection of reducing end biotin (90 µg/mL) was performed over channels A–C, (2,260 min, red arrows, inset). (C) Effect of altering the mole percentage of biotin OEG in the SAM on the amount of streptavidin captured. Three surfaces with SAMs assembled using the indicated mole percentage biotin OEG were prepared. (1 mL, 20 µg/mL) was injected over the surfaces. The traces from each experiment are superimposed to enable comparison. Data for the three measurement channels A–C are the mean of the response in these channels.

The heparin functionalised surfaces could be removed from the instrument and stored at 4 °C in PBS for up to 10 days without any detectable loss of analyte binding.

To determine whether the amount of streptavidin immobilised on the surface could indeed be controlled by changing the percentage thiolated OEG in the SAM, SAMs with three different percentage thiolated biotin OEG were functionalised with streptavidin. The data demonstrate a dependence of the level of streptavidin captured on the SAM and the percentage thiolated biotin OEG in the original ligand mixture used to assemble the SAM (Fig. 1C). This in turn enables the density of biotinylated immobilised ligand to be controlled. In the previous work, maximal areal density of streptavidin was obtained with 0.1% mole/mole thiolated biotin OEG (Migliorini et al., 2014), but in the present work this did not afford anything like full coverage of the surface. That 1% (mole/mole) is likely to do so is supported by several lines of evidence. Supported lipid bilayers on glass require 5% mole/mole biotinylated lipid for full coverage by streptavidin, which is higher than that obtained on the gold surface, likely due to be due to the mobility of the lipids increasing the streptavidin's packing density (Migliorini et al., 2014). OEG is considerably smaller than a lipid and with streptavidin occupying ~23 nm$^2$, then at 1% mole/mole thiolated biotin OEG will be sufficient to provide a streptavidin monolayer. This monolayer will have gaps, simply due to inefficient packing that are the consequence of the thiolated OEG molecules being immobilized on the gold surface.

The heparin functionalised surfaces were then tested for FGF2 binding, as the interaction of this growth factor with heparin has been well characterised, including the measurement of binding kinetics in optical biosensors, e.g., (Delehedde et al., 2002). First the running buffer was changed to PBST. FGF2 (1 mL, 100 nM) was injected over the three measurement channels, previously functionalised with biotin heparin immobilised on streptavidin (Fig. 3A) and the control channel (functionalised solely with streptavidin, Fig. 3A dotted line). There was little response in the control channel indicating that, as previously observed, FGF2 did not bind streptavidin (Rahmoune et al., 1998) or the underlying OEG SAM (Migliorini et al., 2014). In contrast, in the measurement channels there was a significant response (~1.2 nm). When the channels were returned to running buffer, the response was stable. The lack of dissociation (or very slow dissociation) is a hallmark of the rebinding of a dissociated ligand before it can diffuse into the flowing bulk solution (Schuck & Zhao, 2010). Rebinding is particularly acute with interactions possessing a significant electrostatic component, as is the case with protein-sulfated glycosaminoglycan binding, since these have fast association rate constants due to electrostatic steering. Rebinding can be overcome by injection of non-biotinylated ligand, in this case heparin (Sadir, Forest & Lortat-Jacob, 1998). As expected, this resulted in a dissociation curve (Fig. 4A). Washing the surface with 1 mL 2 M NaCl and then 20 mM HCl completely regenerated the surface, as the baseline returned to its original value (Fig. 3A). The second 20 mM HCl regeneration step was included to ensure both full regeneration of the surface and that no analyte remained associated with the fluidics.

Recently, we have described two new assays for the identification of inhibitors of heparan sulfate sulfotransferases (HSSTs) (Byrne et al., 2018b), the enzymes responsible for the sulfation of precursors of heparan sulfate and heparin. These assays can measure

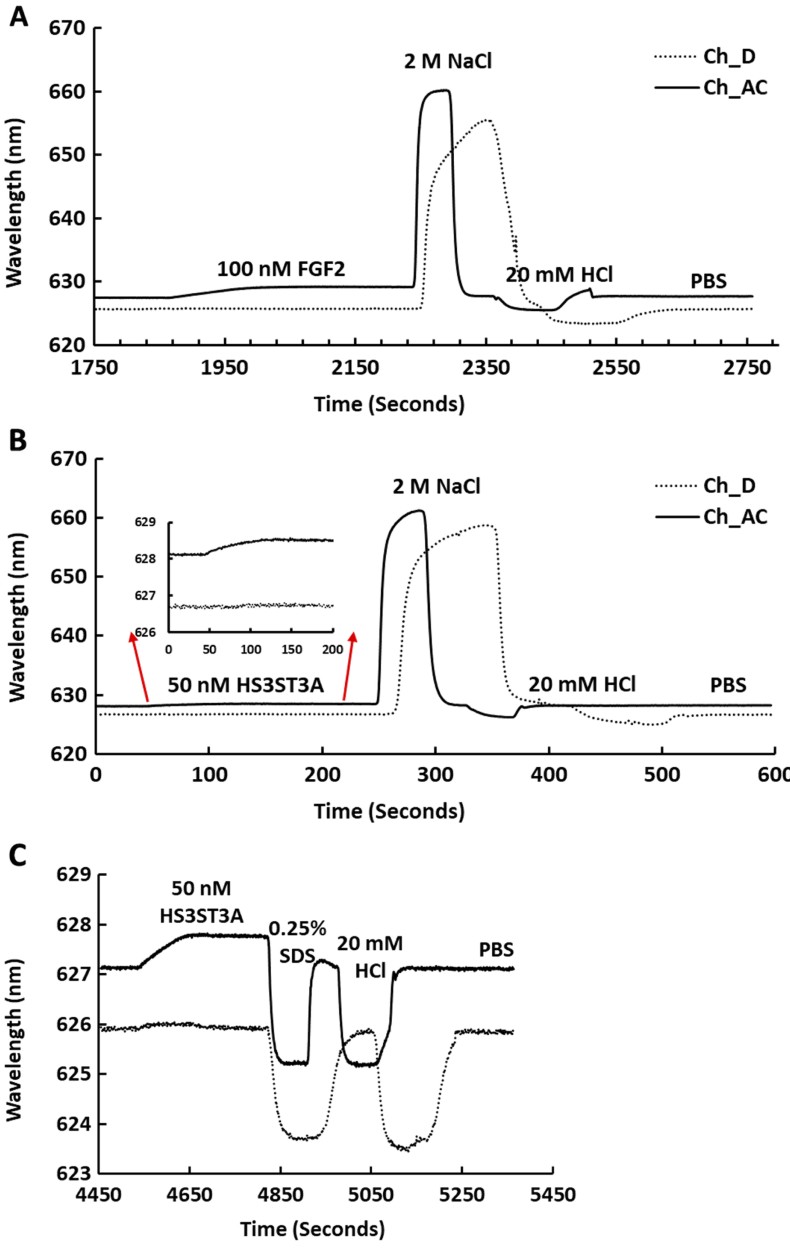

**Figure 3 Binding of FGF2 and GST-HS3ST3A1 to a heparin derivatised surface.** Streptavidin was captured on a 1% (mole/mole) biotin OEG SAM and reducing end biotinylated heparin was in turn captured on channels A–C, leaving channel D as the control for non-specific binding, as in Fig. 2B. The flow rate of PBST over the surface was maintained at 500 μL/min. (A) 1 mL 100 nM FGF2 was injected over channels A–C (biotinylated heparin) and channel D (streptavidin), followed by regeneration of the surface by sequential washes with 1 mL 2 M NaCl and 20 mM HCl. (B) 1 mL 50 nM GST-HS3ST3A1, followed by regeneration with 2 M NaCl and 20 mM HCl. Inset: expanded view of GST-HS3ST1A1 binding. (C) 1 mL 50 nM GST-HS3ST3A1, followed by regeneration with 1mL injection of 0.25% (w/v) SDS and then 20 mM HCl.

whether a compound interacts with a sulfotransferase resulting in a change to its thermal stability or the activity of the sulfotransferase towards a model oligosaccharide substrate. The latter assay can also determine whether the compound is competitive with

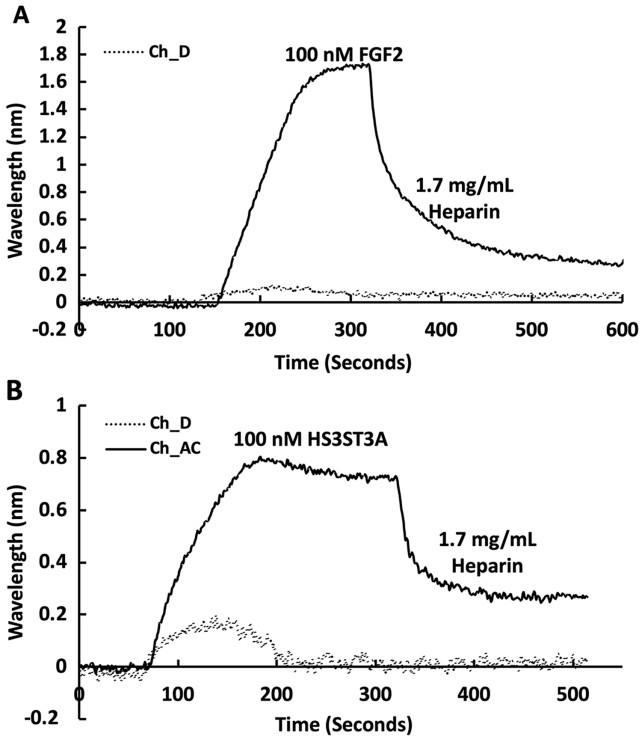

**Figure 4 Dissociation induced by soluble heparin.** Channels A–C on a chip with streptavidin captured on a 1% mole/mole biotin OEG SAM were functionalised with biotinylated heparin. 1 mL (A) 100 mM FGF2 and (B) 100 mM GST-HS3ST3A1 were injected at 500 µL/min. After the surface was returned to running buffer 1 mL heparin in PBS was injected over the surface.

3′-phosphoadenosine-5′-phosphosulfate (PAPS), the universal sulfate donor. However, this assay cannot determine whether a compound is competing with the sugar acceptor, since the sulfation of the sugar acceptor is measured. Consequently, it was of interest to determine whether it would be possible to develop SPR assays to explore the selectivity of a HSST for particular patterns of sulfation of the sugar acceptor, and whether heparin-binding might be used to identify likely acceptor competitors. When 50 nM HS3STA1 was injected over the control, streptavidin functionalised surface, the small response observed returned to baseline as soon as the injection ended, and the surface had been returned to running buffer (Fig. 3B, dotted line). This response is characteristic of a bulk shift, presumably arising from the refractive index of the GST-HS3ST3A1 and its buffer being higher than that of running buffer. Thus, HS3STA3A1 does not interact with streptavidin or the underlying OEG SAM. Injection of GST-HS3ST3A1 over the heparin-derivatised surface resulted in a response of 0.5 nm (Fig. 3B). As observed with FGF2 (Fig. 3A), when the surface returned to running buffer, there was little dissociation (Fig. 3B). Again, dissociation could be induced by the injection of heparin over the surface (Fig. 4B). The heparin-derivatised surface was successfully regenerated with a wash with 2 M NaCl followed by one with 20 mM HCl (Fig. 3B). However, upon multiple injections, a small, but measurable rise in baseline was sometimes observed, indicative of incomplete removal of the bound analyte. Consequently, the surface was thereafter
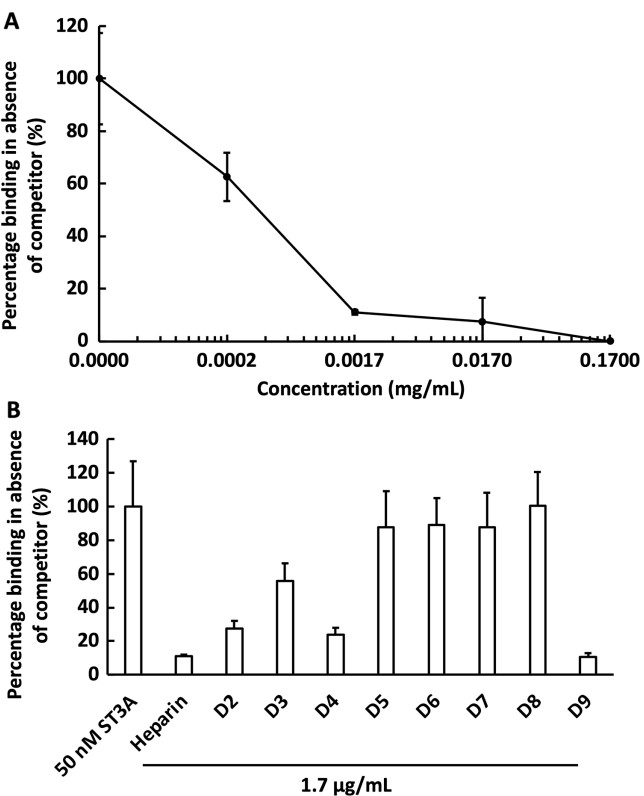

**Figure 5 Competition by heparin derivatives for HS3STA1 binding to immobilised heparin.**
Channels (A–C) on a chip with streptavidin captured on a 1% mole/mole biotin OEG SAM were
functionalised with biotinylated heparin. GST-HS3ST3A1 (50 nM, 1 mL) mixed with increasing con-
centrations of (A) heparin, and (B) 1.7 µg/mL heparin and heparin derivatives (Table 1). Between each
injection the surface was regenerated by a 1 mL injection of 2 M NaCl followed by 1 mL 20 mM HCl.
The response was the difference in nm of signal between the initial baseline in PBST and the baseline
achieved after the surface returned to PBST following the injection of analyte; the percentage binding was
calculated with respect to the response in the absence of added competitor. The data in this figure and in
Fig. 6 were acquired on the same surface. All values were measured by a single injection over the three
measurement channels, so as triplicate technical repeats. However, the control values for 50 nM GST-
HS3ST3A1 were determined independently eight times, twice at the start, then every fourth injection, to
ensure the surface was responding consistently throughout. The control value is, therefore, the mean of
eight independent triplicate repeats and is the same for (A) and (B) and for Fig. 6. Error bars are the
means ± SD.                                                   

regenerated with a wash of 0.25% (w/v) SDS; a second wash with 20 mM HCl was
undertaken to ensure that no residual SDS was present on the fluidics or the surface
(Fig. 3C). These latter data demonstrate that 0.25% (w/v) SDS does not remove bound SA
or damage the OEG SAM, but did afford full regeneration when many injections of the
enzyme were performed.

We next examined the competition for GST-HS3ST3A1 binding to the heparin-
derivatised surface by soluble heparin. GST-HS3ST3A1 (50 nM final concentration) was
pre-mixed with heparin to yield the indicated final concentration. Samples were injected
over the heparin-derivatised surface and then regenerated with 2 M NaCl and 20 mM
HCl prior to the next injection. The data demonstrate dose-dependent inhibition of
binding by soluble heparin with an $IC_{50}$ of under 0.34 µg/ml heparin (Fig. 5A). To identify

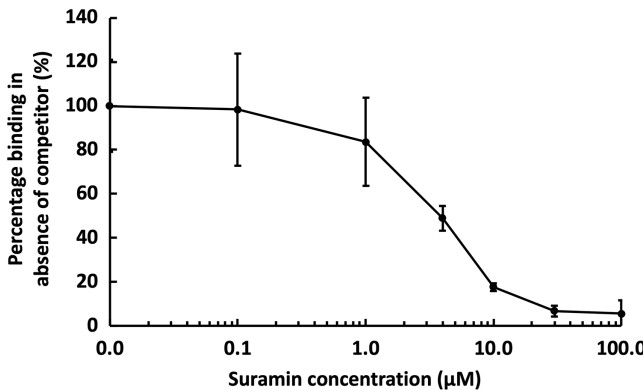

**Figure 6 Competition by suramin for GST-HS3ST3A1 binding to immobilised heparin.** One mL 50 nM GST-HS3ST3A1 in the presence of increasing concentrations of suramin was injected over a heparin functionalised surface. The response measured was the difference in nm of signal between the initial baseline in PBST and the baseline achieved after the surface returned to PBST following the injection of analyte. Results are the mean ± SD of three technical repeats and are expressed as a percentage of the response measured in the absence of suramin, which was determined in eight independent triplicate repeats, and is the same as that in Fig. 5.   

sulfation patterns preferentially recognised by GST-HS3ST3A1, competition assays using a library of modified heparins were performed. Selective removal of sulfate groups to produce the different modified heparins (Table 1) was expected to reduce their inhibition of the binding of the enzyme to the immobilised heparin, as seen previously with other heparin/HS binding proteins (*Li et al., 2016*). Consequently, a final concentration for the heparin derivatives of 1.7 μg/mL was chosen, since with the parent heparin, this provided close to 100% inhibition (Fig. 5A). Selective removal of any one of the sulfate groups on heparin reduced somewhat the effectiveness of the derivative as a competitor, although these different derivatives, D2 to D4, were not equivalent, indicating that effective interaction also involved a degree of conformational compatibility. Thus, heparin derivatives with sulfate at the C2 position of iduronate residue and sulfate at either the C6 or the C2 positions of the glucosamine residue, D2 and D4, respectively, were nearly as successful competitors as the parent heparin (Fig. 5B). In contrast, the heparin with sulfate at C2 and C6 of glucosamine residues, D3, was a less effective inhibitor. These data indicate that GST-HS3ST3A1 has a preference for a polysaccharide containing 2-*O* sulfated iduronate and either 6-*O* or *N*-sulfated glucosamine. Loss of any two, or all three, sulfate groups resulted in heparin derivatives that were less effective competitors (Fig. 5A). Persulfated heparin (D9) was as effective a competitor as the parent heparin, but not more effective, which indicated that simple charge density was not entirely responsible for the interaction. These data are consistent with the observation that GST-HS3ST3A1 is a 'gD' HS 3-*O* sulfotransferase (*Moon et al., 2004*), which sulfates the hydroxyl group at the C3 position of *N*-sulfated glucosamine residues in the context of flanking 2-*O* sulfated iduronate residues, to produce structures that bind the gD protein of herpes simplex virus (*Moon et al., 2004*).

Suramin, one of the compounds identified in a recent screen of inhibitors of HS2ST1 was predicted by modelling to compete at least in part for the sugar acceptor site on

the enzyme (*Byrne et al., 2018b*). This was consistent with the documented ability of suramin to compete with, for example, the binding of FGF2 to heparan sulfate (*Fernig, Rudland & Smith, 1992*). The observation that suramin was also an inhibitor of tyrosylprotein sulfotransferase 1 (*Byrne et al., 2018a*), however, suggested that suramin might be a fairly generic sulfotransferase inhibitor that competed with the universal sulfate donor PAPS. We therefore tested whether suramin would compete for GST-HS3ST3A1 binding to heparin. Suramin was premixed with 50 nM HS3ST3A1 to provide the indicated final concentrations (Fig. 6). The data demonstrate a dose-dependent inhibition of heparin binding by suramin with an $IC_{50}$ of around 5 µM (Fig. 5A). This provides experimental support for the modelling studies presented previously (*Byrne et al., 2018b*). Thus, the heparin derivatised surface also provides the means to identify compounds that compete for the polysaccharide acceptor substrate that are likely to be inhibitors of the HSSTs.

## CONCLUSIONS

The surface assembled here provides a versatile means by which to capture biotinylated ligands and is characterised by a low level of non-specific binding. At each stage of assembly, these surfaces may be stored and, in the case of the final heparin functionalised surface, re-used and are resistant to a range of regeneration methods. The thiolated OEG/thiolated biotin OEG surfaces can be stored at 4 °C in ethanol for at least three months, while streptavidin and the biotin heparin functionalised surfaces can be stored at 4 °C in PBS for at least 10 days, and the latter re-used over a week. Moreover, functionalised surfaces can be plasma cleaned and re-used 2–3 times. Consequently, these surfaces are entirely compatible with the low cost and portability of the instrument and provide an alternative to the peptide surface employed previously *e.g.*, (*Bolduc et al., 2009*; *Bolduc et al., 2011*; *Bolduc, Pelletier & Masson, 2010*). The stability of the heparin derivatised surface makes this an interesting route to the development of generic capture surfaces for 'sandwich' type assays, since many proteins associated with pathological changes (*Nunes et al., 2019*; *Salamat-Miller et al., 2007*) including viral ones bind heparin, a close structural analogue of the naturally occurring cell surface receptor for many viruses, including SARS-CoV-2 (*Kim et al., 2020*; *Liu et al., 2021*; *Mycroft-West et al., 2020*).

## ACKNOWLEDGEMENTS

We thank Richard Nichols, Department of Chemistry, Univeristy of Liverpool for training and access to a plasma cleaner during lockdown.

### Funding

This work was supported by the European Commission FET-OPEN grant [ArrestAD no. 737390] to David G. Fernig and Edwin A. Yates. The funders had no role in study design, data collection and analysis, decision to publish, or preparation of the manuscript.

## Grant Disclosures

The following grant information was disclosed by the authors:
European Commission FET-OPEN: 737390.

## Competing Interests

The authors declare they have no competing interests.

## Author Contributions

- Dunhao Su performed the experiments, analyzed the data, prepared figures and/or tables, authored or reviewed drafts of the paper, and approved the final draft.
- Yong Li performed the experiments, prepared figures and/or tables, authored or reviewed drafts of the paper, and approved the final draft.
- Edwin A. Yates conceived and designed the experiments, performed the experiments, authored or reviewed drafts of the paper, and approved the final draft.
- Mark A. Skidmore conceived and designed the experiments, authored or reviewed drafts of the paper, and approved the final draft.
- Marcelo A. Lima conceived and designed the experiments, authored or reviewed drafts of the paper, and approved the final draft.
- David G. Fernig conceived and designed the experiments, performed the experiments, analyzed the data, authored or reviewed drafts of the paper, and approved the final draft.

## Data Availability

The raw data are available in the Supplemental File.

## Supplemental Information

Supplemental information for this article can be found online at http://dx.doi.org/10.7717/peerj-achem.15#supplemental-information.

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
