# Peer review of "Analysis of protein-heparin interactions using a portable SPR instrument"

_PeerJ Analytical Chemistry, doi:10.7717/peerj-achem.15_

## Round 0.1 · original submission · Minor Revisions

Please carefully address the comments provided by the reviewers.

Reviewer 1 ·

Basic reporting

This organization of the paper is good. The manuscript has been written very clearly in a very unambiguous and professional language. The figures have been appropriately described and labelled and relevant to the content of the article. Sufficient background information has been provided in the literature.

Experimental design

The methodology has been described in sufficient details with thorough investigation and well defined research question.

Validity of the findings

The results have been depicted convincingly with a well stated conclusion.

Reviewer 2 ·

Basic reporting

English is clear
Literature sufficient
Articles structure generally OK, except that the authors go back & forth in citing Figures, particularly Figure 3-6

Experimental design

Original research, well described, relevant and meaningful

Validity of the findings

Supplemental/Raw data was not provided
Replicate information was not clear
Conclusions were clear

Additional comments

In this study, the authors have attempted to provide an optimized method/workflow for passivation and functionalization of a portable SPR instrument to be used in the field. Specifically, the authors use biotin-OEG::OEG in combination with streptavidin to generate the capture surface. The authors then provide examples of how it could be potentially applied in real-world scenarios.
Overall, the experimental design is simple, and the study appears convincing. The study could be published given the following minor comments are addressed:

Did the authors performed replicate injections and only the representative data were provided?

Did the authors test percentage of biotin-OEG over >1%? If yes, could the data be shown? If not, why not?

The authors optimized/prepared surfaces using PBS as the running buffer but then switched to PBST for their FGF2 measurements. Is there a reason for it? If measuring FGF2 in PBST is important, I suggest that the surfaces be prepared in PBST as well.

Why the authors choose 50 nM GST-HS3ST3A1 for Figure 3B and not 100 nM as they did in Figure 4B? This is important for a proper comparison between streptavidin/OEG and heparin binding efficiencies.

In Figure 3C, a side-by-side comparison of SDS/HCl and NaCl/HCl will be useful for comparison.

What is the motivation for using 0.25% SDS in Figure 3C? The authors should explain this in the text as it appears as a surprise!

In Figure 3C, Ch_D dropped below baseline after HCl, why?

Reviewer 3 ·

Basic reporting

This paper described the assembly of a surface of thiolated oleyl ethylene glycol/biotin oleyl ethylene glycol and its functionalisation with streptavidin and reducing end biotinylated heparin for a SPR instrument. Two examples of the analysis of heparin-binding proteins are presented. Overall the structure of the paper is clear, while a number of clarification and modifications are necessary.

The authors don't cite enough references to claim the necessary of using heparin of this research.

The flow rate of the running buffer is 500μL/min, while it is too fast for molecular reaction and cost a lot of reaction reagent in the SPR experiment The authors should explain the reason for choosing this flow rate which is always lower than 50 μL/min in other common SPR instrument.

Why two regeneration steps were conducted, and usually acid or alkali solution can achieve one-step regeneration. It should be more clearly discussed.

Experimental design

no comment

Validity of the findings

no comment

Additional comments

no comment

---

## Round 0.2 · accepted · Accept

The authors have addressed the reviewers' comments.

Reviewer 2 ·

Basic reporting

No negative remarks

Experimental design

No negative remarks

Validity of the findings

No negative remarks

Additional comments

No negative remarks